# Direct Detection of Antibacterial-Producing Soil Isolates Utilizing a Novel High-Throughput Screening Assay

**DOI:** 10.3390/microorganisms10112235

**Published:** 2022-11-11

**Authors:** Wikus Ernst Laubscher, Marina Rautenbach

**Affiliations:** BioPep Peptide Group, Department of Biochemistry, University of Stellenbosch, Stellenbosch 7600, South Africa

**Keywords:** antibiotic discovery, soil organisms, bacterial peptide producers, high throughput assay, bioluminescence

## Abstract

The ever-increasing global threat of common infections developing resistance to current therapeutics is rapidly accelerating the onset of a primitive post-antibiotic era in medicine. The prevention of further antimicrobial resistance development is unlikely due to the continued misuse of antibiotics, augmented by the lack of discovery of novel antibiotics. Screening large libraries of synthetic compounds have yet to offer effective replacements for current antibiotics. Due to historical successes, discovery from large and diverse natural sources and, more specifically, environmental bacteria, may still yield novel alternative antibiotics. However, the process of antibiotic discovery from natural sources is laborious and time-consuming as a result of outdated methodologies. Therefore, we have developed a simple and rapid preliminary screening assay to identify antibacterial-producing bacteria from natural sources. In brief, the assay utilizes the presence or absence of luminescence in bioluminescent reporter bacteria and test bacterium co-cultures in a 96-well plate format to determine the absence or presence of antibacterial compound production. Our assay, called the bioluminescent simultaneous antagonism (BSLA) assay, can accurately distinguish between known antibacterial-producing and non-producing test bacteria. The BSLA assay was validated by screening 264 unknown soil isolates which resulted in the identification of 10 antibacterial-producing isolates, effectively decreasing the pool of isolates for downstream analysis by 96%. By design, the assay is simple and requires only general laboratory equipment; however, we have shown that the assay can be scaled to automated high-throughput screening systems. Taken together, the BSLA assay allows for the rapid pre-screening of unknown bacterial isolates which, when coupled with innovative downstream dereplication and identification technologies, can effectively fast-track antimicrobial discovery.

## 1. Introduction

There is no doubt that antibiotics have been one of the most valuable medical discoveries to date. Today, what appears to be a trivial infection, such as an infected wound, can in most cases be effectively treated by antibiotics while a century ago there was a big chance of septicemia and possible death. Moreover, the risk of medical surgeries has substantially decreased, resulting in the rapid innovation of more advanced novel treatments in several unrelated medical disciplines [1]. However, throughout the last several decades, the efficacy of these miracle drugs has been decreasing. The ever-increasing development of antimicrobial resistance (AMR) is threatening the onset of a “post-antibiotic era” with severe health and financial implications [2]. The COVID-19 pandemic impeded the antimicrobial resistance data collection; however, the available data indicated an alarming 15% increase in hospitalized patients with resistant infections and associated deaths from 2019 to 2020 [3]. There was also about 80% of patients with COVID-19 who received an antibiotic, which signified the escalated use of antibiotic utilization and the potential escalation of AMR during this pandemic [3]. It has been estimated that annually in the USA alone, AMR results in 2.8 million infections which are directly linked to more than 35,000 deaths and a financial burden of at least USD 55 billion [3,4]. 

Antimicrobial resistance is by no means a new phenomenon. As early as 1924, predating the discovery of penicillin, antimicrobial resistance was reported against the antimicrobial such as Salvarsan, introduced in 1910 for the treatment of syphilis [5,6]. Many other reports on the resistance towards Salvarsan soon followed [7,8]. Similarly, 50% of *Staphylococcus aureus* infections in Britain were reported to be resistant to penicillin by 1949, only five years after its introduction to the market in 1944 [9]. At the beginning of the 1940s, the work of Selman Waksman resulted in the start of what is now known as the “golden era” of antibiotics [10]. His systematic use of an agar overlay assay to detect antibiotic activity from soil microbe extracts resulted in the discovery of the macrolide, aminoglycoside, glycopeptide, and amphenicol classes of antibiotics [11,12]. These are most of the major classes of antibiotics, most of which are still in use today. The seemingly unlimited availability of novel antibiotics, therefore, made the treatment of antimicrobial resistance simple. As soon as resistance developed against one antibiotic, a novel antibiotic is available to take its place. Unfortunately, the “golden era” of antibiotics ended in the late 1960s, as the Waksman pipeline of discovery started to run dry due to the constant rediscovery of old antibiotics [10]. Furthermore, the advent of various chemotherapies in unrelated medical disciplines resulted in a decreased interest in antibiotic discovery and development.

Combinatorial chemistry now provides the ability to rapidly synthesize thousands of compounds that can be rapidly screened against targets in high-throughput screening (HTS) pipelines aided by robotics. Furthermore, the advent of genomics, proteomics, and systems biology can identify probable antibiotic targets, therefore guiding combinatorial synthesis. However, these promising modern technologies have yielded disappointing results in identifying novel classes of antibiotics, mostly due to the poor permeability of synthesized compounds over the bacterial cell wall and membrane. This resulted in many suggesting the resurrection of antibiotic discovery from natural sources owing to the major success of Waksman. This might seem as if we are taking a step back; however, today we have access to novel technologies to both access and study previously inaccessible natural environments. As an example, genomics revealed that traditionally we have only been able to study about 1% of bacteria in the soil microbiome [13]. Novel culturing methods are constantly being developed to culture previously unculturable environmental microbes [14]. Most notable was the use of the Ichip for the in situ cultivation of soil microbes resulting in the discovery of the antibiotic teixobactin from the previously unculturable bacterium *Eleftheria terrae* [15,16]. Furthermore, several untouched environments have yet to receive full attention in the field of antibiotics, such as extreme and marine environments. Coupled with its historical success, there is thus ample incentive to revisit antibiotic screening from natural sources.

Although successful, the methods used by Waksman and many others at the time were tedious and time-consuming [17]. These methods are still used today due to the lack of natural product screening methodology development especially compared to their synthetic counterparts. Therefore, we developed a novel up-to-date assay for the rapid early identification of antibacterial-producing soil bacteria called the bioluminescent simultaneous antagonism (BLSA) assay (Figure 1). The BLSA assay layout is simple and requires only general laboratory equipment and visual and/or basic analyses to interpret the results. An important novelty of our BSLA assay is that it can be scaled up to be incorporated into automated liquid handling systems for HTS. The BLSA assay entails the co-culturing of a test bacteria together with an autonomous bioluminescent reporter bacterium constitutively expressing an integrated *luxCDABE* gene cassette from *Photorhabdus luminescens*. The *luxAB* genes encode the α and ß subunits of luciferase which catalyzes the oxidation of reduced flavin mononucleotide (FMNH_2_) and fatty aldehyde synthesized by the fatty acid reductase complex encoded by the *luxCDE* genes, resulting in the emission of blue-green light [18]. Upon cell death, the concentration of FMNH_2_ drops almost instantaneously, allowing for accurate monitoring of cell death by loss of luminescence [19,20,21,22]. Therefore, reporter bacterium bioluminescence is utilized for cell viability assessment to indicate the presence or absence of antibacterial compounds. 

In this study, we present the assay layout optimization and viability using bioluminescent *Escherichia coli* and *Staphylococcus aureus* as representatives of Gram-negative and Gram-positive bacterial targets. However, the BSLA assay is easily adaptable and can be used with any luminescent target, including fungi as reporter organisms, or even luminescent cancer cells and blood-borne parasites, such as the malaria parasites (*Plasmodium* spp.) and parasites causing sleeping sickness (*Trypanosoma* spp.).

## 2. Materials and Methods

### 2.1. Bacterial Strains

The bacterial strains *Aneurinibacillus migulanus* ATCC9999, *Bacillus subtilis* ATCC21332, *Brevibacillus parabrevis* ATCC10068, *Brevibacillus parabrevis* ATCC8185, Escherichia coli K12 (ATCC10798), and *Pseudomonas aeruginosa* ATCC27853 were obtained from the American Type Culture Collection (Manassas, VA, USA). The Bacillus Genetic Stock Center (Ohio State University, OH, USA) supplied *Bacillus subtilis* OKB105, *Bacillus subtilis* OKB120, and *Bacillus subtilis* OKB168. The National Collection of Type Cultures (Salisbury, UK) supplied *Micrococcus luteus* NCTC8340. *Escherichia coli* Xen14 and *Staphylococcus aureus* Xen29 were obtained from Perkin Elmer (Johannesburg, South Africa). Detailed descriptions of these luminescent bacterial strains can be obtained from https://www.perkinelmer.com/uk (accessed on 3 November 2022). *Bacillus licheniformis* LB5 and *Brevibacillus laterosporus* LB4 were from the curated culture collection of the BioPep^TM^ Group at Stellenbosch University (Stellenbosch, South Africa). 

### 2.2. Bacterial Growth

During this study, the bacterial strains *A. migulanus* ATCC9999, *B. licheniformis* LB5, *B. subtilis* ATCC21332, *B. subtilis* OKB105, *B. subtilis* OKB120, *B. subtilis* OKB168, *Br. latero-sporus* LB4, *Br. parabrevis* ATCC10068, *Br. parabrevis* ATCC8185, *E. coli* K12, *M. luteus* NCTC8340, and *P. aeruginosa* ATCC27853 are collectively referred to as test bacteria. These strains were streaked from freezer stocks at −80 °C on LB agar (LA: 10 g/L tryptone, 5 g/L yeast extract, 10 g/L sodium chloride, and 15 g/L agar) and incubated at 30 °C for 48 h before use in downstream assays.

*E. coli* Xen14 and *S. aureus* Xen29, collectively referred to as reporter bacteria, were streaked onto LB agar supplemented with 30 µg/mL or 200 µg/mL kanamycin, respectively, and incubated at 37 °C for 48 h. Subsequently, 3 mL of LB broth (LB: 10 g/L tryptone, 5 g/L yeast extract, and 10 g/L sodium chloride) in 14 mL falcon round bottom test tubes (Corning, New York, NY, USA) were inoculated with a single colony of the reporter bacteria and incubated at 37 °C for 16 h while shaking. Cells were sub-cultured (1%) in LB and incubated at 37 °C until the mid-log phase, equivalent to an OD_600nm_ of 0.4, before being used in downstream assays.

### 2.3. Bioluminescent Simultaneous Antagonism Assay

Prior to the BLSA assay, sterile non-treated black 96-well plates and two standard sterile 96-well plates per black plate (Thermo Fisher Scientific, Roskilde, Denmark) were prepared by the addition of a 100 µL layer of semi-solid growth medium referred to as the test medium. Following solidification, plates were stored at 4 °C for no more than 7 days before use. The black BSLA assay plates and duplicate 96-well plates were subsequently inoculated with test bacteria, grown as described previously by transferring a single colony per well from streak plates. Colony transfer was conducted by inserting the tip of a 200 µL pipette tip into a colony on an agar streak plate followed by gentle dabbing of the tip of the pipette tip onto the surface of the agar in the well. Wells used as controls were not inoculated. The 96-well plates were covered with plate cover and incubated for 7 days at 30 °C to allow the formation of established/confluent colonies covering most of the agar surface in the well. The humidity in the incubator was kept at 85% to 95% to reduce evaporation during extended incubation periods. The reporter bacteria at the mid-log phase, grown as described above, were diluted to 5 × 10^7^ colony-forming units per mL (CFU/mL) in LB before being diluted 100-fold to a final concentration of 5 × 10^5^ CFU/mL in the final growth medium, hereafter referred to as the reporter medium. A 100 µL aliquot of culture suspension was transferred to black plate wells on top of test bacterium colonies. Wells without test/sample bacteria or reporter bacteria were used as the positive control. Wells with reporter bacteria but no test bacteria were used as negative controls. Following incubation at 30 °C under high humidity as described earlier, reporter bacterium luminescence was determined as counts per second using a one second read time per well on a Tecan Spark 10M multimode microplate reader (Tecan, Männedorf, CH). Plates were also visualized on a myECL imager (Thermo Fisher Scientific) using the chemiluminescent mode. 

Active hits, for further workup from duplicate plates, were defined as positive hit bacteria resulting in a reporter bacterium luminescence below the antibacterial threshold (AbT) defined in Equation (1).
(1)$$AbT=\mu_{pos}+4\sigma_{neg}$$

Raw data from separate .xlsx files generated by the Tecan Spark 10 M multimode microplate reader were combined into a single dataset and transformed into the desired structure using the dplyr and tidyr packages from the tidyverse collection of packages inside the R statistical software [23,24]. The ggplot2 package from the tidyverse collection of packages was used for creating plots [24].

Refer to Figure 1 for a schematic representation of the BSLA assay.

### 2.4. Mass Spectrometry of Test Culture Extracts

Bacteria were grown in parallel with the bioluminescent simultaneous antagonism assay described above on LB agar in the duplicate 96-well plates at 30 °C for 3, 5, or 7 days before being extracted with 200 µL methanol. Extracts were subjected to direct injection electron spray mass spectrometry on a Waters Synapt G2 mass spectrometer. The cone and capillary voltage of the ionization source was set to 15 V and 2.5 kV, respectively. Nitrogen was used as desolvation gas at 650 L/hour and the desolvation temperature was set to 275 °C. Extracts (3 µL) were injected and eluted in the absence of a column using an isocratic gradient of 1:1 acetonitrile and 0.1% formic at 0.25 mL/min. Data were collected for 1 min by scanning over an *m/z* range of 200 to 2000 in positive mode. All spectra per injection were combined in the MassLynx V4.1 software and deconvoluted using the MaxEnt3 algorithm included in the MassLynx V4.1 software. Refer to Appendix A for mass spectrometry results on the culture extracts

### 2.5. Test Bacterium Activity

The activity of the twelve test bacteria were confirmed using the traditional simultaneous antagonism assay. Molten LA at 50 °C was inoculated with reporter bacteria at the mid-log phase to a final concentration of 5 × 10^5^ CFU/mL. The reporter bacterium suspensions were subsequently poured (5 mL) onto LA-containing Petri dishes and allowed to solidify. Immediately after solidification, test bacteria were introduced as a spot on the reporter bacterium layer, using a sterile 200 µL pipette tip to transfer a single colony from a streak plate (prepared as described above). Following 24 h of incubation at 30 °C, plates were visualized using a myECL imager. The activity was defined as a clear zone surrounding a test bacterium colony.

The activity of the twelve test bacteria were determined by the BLSA assay, as described above, and compared to the expected activity as confirmed by the simultaneous growth antagonism assay to determine assay performance. LA was used as a test medium in the assay for the selected colony growth. Several growth media were investigated for the two model reporter (target) bacteria which included LA, LB, cation-adjusted Mueller Hinton broth (MHB: 22 g/L Mueller Hinton broth 2), MH-agar (MHA: MHB with 15 g/L agar), nutrient broth (NB: 13 g/L nutrient broth no. 3), N-agar (NA: NB with 15 g/L agar), tryptic soy broth (TSB: 30 g/L tryptic soy broth), and TS-agar (TSA: TSB with 15 g/L) growth media. Luminescence was measured and plates were visualized following 24 and 48 h of incubation. Each test contained three technical repeats and was repeated for a total of three biological repeats. Wells containing only the luminescent reporter bacteria were used as negative controls. Positive controls for 100% inhibition contained no test culture or reporter bacteria.

Raw data were transformed and plotted as described before using the dplyr, tidyr, and ggplot2 packages in the R statistical software [23,24]. Furthermore, one-way ANOVA followed by the Tukey posthoc test was performed using the R statistical software [23]

### 2.6. BLSA Assay pH Resistance

The pH of molten LB agar (1.5% *w/v*) was adjusted to a pH of 7, 6, 5, 4, and 3, respectively, with lactic acid at 60 °C. The molten agar was transferred to the wells of a black 96-well plate (100 µL) and allowed to set which constituted the test medium. The plates were stored at 4 °C for no more than 6 h before use. Reporter bacterium culture suspensions at the mid-log phase, prepared as described previously, were diluted in the NB reporter medium to a final concentration of 5 × 10^5^ CFU/mL. Plates containing the pH-adjusted test medium received 100 µL of the reporter bacterium culture suspension followed by incubation at 30 °C under high humidity. Wells which received reporter medium without the addition of reporter bacteria were used as controls. Reporter bacterium luminescence was recorded as mentioned above following 24 and 48 h of incubation. 

Plotting was conducted using the ggplot2 package following data restructuring using the dplyr and tidyr R packages from the tidyverse collection of packages in the R statistical software [23,24].

### 2.7. Assay Quality Statistics

Reporter bacterium culture suspensions were prepared as above to a final concentration of 5 × 10^5^ CFU/mL in the desired broth growth medium. A 100 µL of culture suspension was transferred to columns one to six of a sterile black 96-well plate containing a base layer of LB agar, prepared as described above (Section 2.3). Columns seven to twelve on the plate received only growth medium without the addition of reporter bacterium. Plates were incubated at 30 °C under high humidity and luminescence was measured as described above at 24 and 48 h. Wells containing reporter bacteria were used as the negative control; conversely, positive controls were represented by wells containing only reporter medium. A total of three biological repeats were conducted on separate plates. The Z’-factor, signal-to-noise (S/R) ratio, signal-to-background (S/B) ratio, and coefficient of variance (CV) were determined by Equations (2)–(5) to assess assay quality and robustness in the R statistical software [25,26]. In the equations below, µ_max_ and µ_min_ represent the mean of the maximum (negative control) and minimum (positive control) signals, respectively. The standard deviation of the respective signals is represented by σ_max_ and σ_min_.

All quality statistics were calculated in the R statistical software following raw data preparation using the dplyr and tidyr packages from the tidyverse collection of packages [23,24]. Plots were subsequently created using the ggplot2 R package [24].
(2)$$Z^{\prime }=\frac{\left(3\sigma_{max}+3\sigma_{min}\right)}{\left|\mu_{max}-\mu_{min}\right|}$$
(3)$$S/N=\frac{\left|\mu_{max}-\mu_{min}\right|}{\sigma_{min}}$$
(4)$$S/B=\frac{\mu_{max}}{\mu_{min}}$$
(5)$$CV_{max}=100\left(\frac{\sigma_{max}}{\mu_{max}}\right)$$

### 2.8. Limit of Detection

Reporter bacteria at the mid-log phase were prepared as previously mentioned and diluted to 5 × 10^5^ CFU/mL in NB growth medium. Subsequently, 200 µL of the culture suspensions were transferred to the first nine wells in the first row of a sterile black 96-well plate. The remaining three wells received only NB growth medium used as control. The plate was incubated for 24 h at 30 °C under high humidity. Following incubation, a 10× dilution range of the cultures was made on the same plate in the subsequent rows. Dilutions close to the AbT value were transferred to a new black 96-well plate and subsequently double-diluted to increase data points close to the limit of detection (LOD). Luminescence was measured as described before. The CFU/mL of the culture was subsequently determined by spread plating of the dilution series. The LOD was defined as the bacterial concentration (CFU/mL) at AbT. A linear regression model was fitted to the linear part of log-transformed data to calculate the LOD defined as the intercept at AbT. In total, three biological repeats were conducted on separate plates. 

Raw data were restructured and transformed using the dplyr and tidyr R packages followed by linear model fitting using the tidymodels packages in the R statistical software [23,24,27].

### 2.9. Soil Sample Screening

Soil sampling was conducted by removing the top 2 cm of soil and transferring the soil up to 10 cm deep into a sterile 50 mL centrifuge tube. A 1.0 g amount of the soil was suspended in a 10 mL PBS buffer (8 g/L NaCl, 0.2 g/L KCl, 1.42 g/L Na_2_HPO_4_, and 0.24 g/L KH_2_PO_4_ adjusted to pH 7.4 with HCl) by vigorous vortexing. The suspension was left for 5 min at room temperature to allow soil particles to settle out. A 10× dilution range of the supernatant was subsequently made in PBS to a final dilution factor of 10^6^. Spread plates from the PBS supernatant dilutions were made on LA plates, followed by incubation at 30 ºC until clear single colonies were visible (±72 h). Single colonies with different morphologies were selected at random from the spread plates. Each selected colony was transferred with a sterile 200 µL pipette tip to the previously prepared 96-well plates, as described above. The first column of plates was not inoculated as it was used as positive/no reporter growth (rows A to D) and negative/reporter growth controls (rows E to H). Each colony was transferred to three different plates per reporter bacterium (added in the broth after 7 days of growth on the agar of transferred colonies) resulting in three biological repeats. Furthermore, in total, 264 randomly selected colonies were transferred to three different plates per repeat. An extra plate of each was inoculated to keep stock of active isolates. The BLSA assay was completed as described before using NB as the reporter medium. Luminescent measurements were taken after 24 h of incubation. Active isolates were defined as isolates that decreased reporter bacterium luminescence below the AbT of 43 counts/s. The activity of active isolates was subsequently confirmed by the simultaneous antagonism assay, as described above.

Raw data were combined and restructured as described before for the BLSA assay using the dplyr and tidyr packages from the tidyverse collection of R statistical software packages [23,24]. Plotting was subsequently conducted using the ggplot2 package [24].

## 3. Results

### 3.1. BLSA Assay Performance against Known Test Bacteria

Our prototype BLSA assay was used to determine the antimicrobial producer activity of 12 known test bacteria to determine assay accuracy and performance (Table 1) on the grounds of antimicrobial producer status, type of bacterium, and availability. Eight test bacteria have known production of antimicrobial compounds, specifically antimicrobial peptides, and one is known to produce rhamnolipids. Two of these bacteria were chosen to have low production and high production of the surfactin lipopeptide complex, respectively. Three of the bacteria were chosen as models to represent the non-production of any known antimicrobial compounds. 

Each of the 12 test bacteria were grown in separate wells in 96-well plates containing LA growth medium for 7 days. The BLSA assay is aimed at screening a diverse array of environmental bacteria, which will result in the simultaneous screening of both slow-growing and fast-growing bacterial strains. As such, an incubation period of 7 days was chosen to provide the majority of cultures, which will be encountered in practice, with enough time to establish colonies and produce antimicrobial compounds. What is important is that there are antimicrobial compounds in/on the agar when the luminescent reporter organism is added on top of the culture. The long incubation time will result in some cell death of fast-growing test bacteria, sporulation of many soil organisms, and slow-growing persister cells. This will be combined with already produced antimicrobial compounds in the well and result in the inhibition of target bacterial cells. This would be a true positive or hit bacterium for good antimicrobial production, while false negatives are normally found when the culture fails to grow or grow and fail to produce antimicrobial compounds.

The antimicrobial peptide production and identification of antimicrobial compounds were confirmed in the duplicate cultures of the 12 test bacteria in a parallel fashion. This was performed by extracting the test bacterium cultures in methanol and the analysis of extracts by high-resolution mass spectrometry (Appendix A). Subsequently, suspensions of reporter bacteria *E. coli* Xen14 (model Gram-negative target) and *S. aureus* Xen29 (model Gram-positive target) were added to the wells, either in agar or in broth, and reporter bacterium luminescence was measured and visualized following 24 and 48 h of incubation. Several wells containing no test bacteria received either growth medium with or without reporter bacterium inoculation used as positive and negative controls, respectively. A variety of growth media were also used in reporter bacterium suspension preparation to investigate the effects of growth medium on assay performance. 

Focusing on only the BLSA assay controls, it was clear that the assay gave a good distinction between the negative and positive controls ranging between three to five orders of magnitude (Figure 2). The variation in reporter luminescence was most notably due to growth medium solidity, especially for *S. aureus* where broth medium resulted in approximately one order of magnitude decrease in luminescence versus that in agar. Conversely, the broth medium increased the luminescence of *E. coli,* correlating with lower variability. Compared to growth medium solidity, growth medium composition did not result in notable differences in reporter bacterium luminescence, except for TSA which resulted in increased luminescence as compared to its semi-solid counterparts. Increased incubation times generally resulted in a decrease in reporter bacterium luminescence.

It has previously been suggested that this decrease is due to the depletion of available FMNH_2_ during the stationary phase [28]. Interestingly, *S. aureus* displayed similar luminescence at both 24 and 48 h of incubation in LB, NB, and MHB growth media. Unlike the negative controls (growth controls or no inhibition), positive control luminescence (blank or full inhibition) was unaffected by time, growth medium composition, or solidity. This is not surprising, as the absence of the luminescent reporter bacteria metabolism and, therefore, the absence of light is unaffected by other well contents. Therefore, positive control luminescence variability is only the consequence of instrument noise. This was further affirmed by our observation that even the measurement of a clean 96-well plate will result in the same luminescence background variability as compared to plates containing only a growth medium (results not shown). 

The co-culture nature of the assay adds additional complexity generally not present in other activity assays, most notably nutritional competition, variable antibacterial compound production, and varying growth rates of test versus reporter bacteria. This affected the luminescence of reporter bacteria in co-culture, unpredictably resulting in either decreased or even increased luminescence likely due to cellular stress, as compared to controls (Figure 3). Decreasing luminescence, therefore, should not be considered the only metric for the selection of antibacterial-producing test bacteria. However, the assay can be used to provide a binary yes or no answer as to whether reporter luminescence and, therefore, metabolism is detected or not, which in the case of the latter is inferred here as antibacterial compound production by the test bacterium. Figure 3 displays this binary nature clearly due to the separation between the luminescent distribution of reporter bacteria in co-culture with the 12 test bacteria that do and do not produce antibacterial compounds. The detailed results of the 12 test bacteria from which Figure 3 was constructed are provided in Appendix A. Although some of the known producer bacteria used to test the assay resulted in high luminescence, these observations were infrequent and were most likely due to the test bacterium either not producing or producing its respective antibacterial compound in too low quantities. The difference that antimicrobial production level has on the detection is clearly exhibited by the difference between the two surfactin producers in the test panel. *B. subtilis* ATCC21332, a wild type surfactin producer, is not a good producer and it does not always reliably produce surfactin. Compared to *B. subtilis* OKB105, a genetically manipulated high surfactin producer, *B. subtilis* ATCC21332 production was too low to exert inhibition above the selected cut-off (Table 1, refer to Figure 4). This highlighted that the BSLA assay was not only selected for antimicrobial producer bacteria, but specifically for good producers. Alternatively, the producer bacterium can itself produce bioluminescent pigments, but such bacteria are rare. To this effect, the assay could lend itself to some false-negative hits. 

Accurate binary classification/selection of antibiotic/antimicrobial producers requires a set threshold, which can be complicated to determine especially where the separation between classes is narrow, such as in the case here. We empirically investigated several metrics and found that a threshold set to four-fold the standard deviation plus the mean of negative controls was suitable. We here refer to this threshold as the antibacterial threshold (AbT). Due to the similarity of negative controls at various time points and in different growth media, as mentioned earlier, the AbT was calculated from all negative controls in the experiments to increase statistical power. This resulted in an AbT of 43 counts/s, which explained most of the variation in producing test bacteria and almost all the variation in non-producing bacteria. Increasing the threshold would result in increased false positive hits with little gain to true positive hits.

To determine the misclassification rate of the assay, test bacteria were classified as either producing or not producing antibacterial compounds according to the measured luminescence in the BLSA assay. Therefore, test bacteria resulting in reporter luminescence less than or equal to the AbT were classified as producers. Conversely, non-producers were classified as test bacteria resulting in a reporter luminescence greater than the AbT. Observed classifications were subsequently compared to the expected activity as determined by the traditional simultaneous antagonism assay (Table 1) to determine the misclassification rate.

Figure 4 displays the misclassification rates observed for the reporter bacteria *E. coli* Xen14 and *S. aureus* Xen29 following 24 and 48 h of incubation in a variety of reporter media. The detailed results, from which Figure 4 was constructed, are given in Appendix A. Considering all conditions and both the reporter bacteria, the assay resulted in a misclassification rate of only 8.4%. This is a testament to the robustness of the BSLA assay to varying growth conditions. 

Furthermore, these results show that increased incubation time generally resulted in decreased *E. coli* misclassification rates, with the exception of the assays utilizing MHA as a reporter medium. Increased incubation time affected the misclassification rate of *S*. *aureus* to a lesser extent, resulting in a decrease only for NB and MHB reporter media. There was little difference between the misclassification of *E. coli* activity when comparing semi-solid and broth reporter media. However, a large discrepancy was observed between semi-solid and broth reporter media in the misclassification rates of activity toward *S. aureus*. 

Similar to the negative controls (Figure 2), no clear optimal reporter medium could be distinguished, apart from broth media performing generally better (Figure 4). Therefore, any of the broth compositions used in this study can serve as a reporter medium in the BLSA assay. However, NB consistently, although not significantly, resulted in the lowest misclassification rate for both *E. coli* and *S. aureus*. Figure 5 displays the log_10_ luminescence of *E. coli* Xen14 and *S. aureus* Xen29 in the NB reporter medium following a 24 and 48 h test bacterium co-culture challenge. These results show a good distinction between producing and non-/weak-producing test bacteria. 

Misclassification was observed as some false-negative classification at 24 h for the test bacterium *A. migulanus* against *E. coli*, and *B. subtilis* OKB120 against *S. aureus*. The weak surfactin producer, *B. subtilis* ATCC 23312, was consistently classified as a non-producer in the BSLA assay, both at 24 h and 48 h. This demonstrates the limitation of the assay towards early false-negative errors, as discussed earlier, and the strength of distinguishing between weak and good producers at 48 h. It should also be noted that the luminescence of *S. aureus* when co-cultured with *E. coli* K12 was close to the AbT. This could be due to the rapid growth rate of *E. coli* K12 which increases nutritional competition, therefore resulting in decreased luminescence. Due to the low misclassification rate in the NB reporter medium, further analysis was performed using only NB as the reporter medium.

Upon closer visual observation, it became apparent why the semi-solid reporter media showed higher variability and misclassification (refer to Figure 4). This type of reporter media resulted in a halo on the edges of the well in some cases where the test bacterium was a potent antimicrobial producer (Figure 6). This halo effect is clearly visible in Figure 6B for the test bacteria *Br. Parabrevis* ATCC8185 and ATCC10068. The effect was attributed to the greater hydrophobicity of the active antimicrobials produced by these strains, namely tyrothricin, resulting in decreased diffusion throughout the semi-solid reporter medium in the well, therefore resulting in the survival of reporter bacterium on the well’s edge. This is an undesirable effect and, therefore, we recommend against using a semi-solid reporter medium.

### 3.2. pH Stability

Some false positive results from challenge bacteria can be due to acid production, such as what would be expected from lactic acid bacteria, etc. To test the effect of pH on the BLSA assay, the luminescence of the reporter bacteria in the NB reporter medium was recorded and the LA test medium pH was adjusted with lactic acid to a pH of 7, 6, 5, 4, and 3. In Figure 7, it can be observed that a test medium pH of 3 resulted in the reduction of the reporter bacterium luminescence below the AbT for both *E. coli* and *S. aureus*. 

Furthermore, luminescence did not increase after 24 h of incubation. Therefore, we assumed reporter bacterium death at a pH equal to or less than 3. Interestingly, the luminescence of both reporter bacteria decreased as pH decreased after 24 h of incubation. However, at 48 h of incubation, the luminescence at a lower pH returned to that of the luminescence at pH 7. The exception was *S. aureus* at a pH of 4, which did not return to the baseline luminescence. However, with longer incubation times, the luminescence might increase back to baseline.

### 3.3. Assay Quality and Robustness

To test assay quality and robustness, the dynamic range and variability in the luminescence of positive and negative controls were investigated in triplicate at 24 and 48 h. The most widely adopted measure of HTS assay quality assessment is the Z’-factor (eq. (2) defined by Zhang et al. [25]. The Z’-factor is a dimensionless coefficient ranging from −∞ to 1 which considers both the variation and dynamic range of HTS signal measurements. HTS assays can be rated by the Z’-factor as either an ideal assay ($Z^{\prime }=1$), an excellent assay ($0.5\leq Z^{\prime }<1$), or a double assay, where ($0<Z^{\prime }<0.5$), binary or “yes/no”-type assay ($Z^{\prime }=0$), or an unusable assay ($Z^{\prime }<0$), as suggested by Zhang et al. [25]. To further investigate dynamic range, the signal-to-noise (S/N) and signal-to-background (S/B) were also calculated, which should be less than ten and two, respectively, for an acceptable assay [26,29]. Finally, the coefficient of variation of the maximum signal (CV_max_) was used to indicate intra-plate variation which should generally be below 15%.

Both test bacteria resulted in a Z’-factor greater than 0.5 at 24; however, at 48 h the Z’-factor for the *E. coli* Xen14 test bacterium decreased to less than 0.2, unlike *S. aureus* which stayed consistent (Figure 8A,B). This decrease is most likely due to the intense luminescence from *E. coli* Xen14; thus, a relatively small variation in cell counts can have a large impact on variation and coincidently on the Z’-factor. Due to the binary design of the BLSA assay, a Z’-factor greater than zero is still acceptable, as suggested by Zhang et al. [25]. Therefore, the BLSA assay is viable for both test bacteria at either 24 or 48 h of incubation according to the Z’-factor. 

The variation in *E. coli* Xen14 luminescence is further demonstrated by the CV_max_, which was determined to be 16% at 24 h and 28% at 48 h. At 24 h, the CV_max_ for *E. coli* Xen14 is only slightly higher than the suggested threshold of 15%; however, at 48 h, intra-plate variability is almost double the suggested threshold. Conversely, *S. aureus* Xen29 had acceptable intra-plate variation at both 24 and 48 h of approximately 10%. 

The BLSA was found to be extremely sensitive with S/N and S/B ratios several orders of magnitude greater than suggested thresholds (Figure 8C,D). *E. coli* Xen14 had almost two orders of magnitude greater sensitivity as compared to *S. aureus* Xen29. The greater sensitivity also directly translated to a better LOD, where the minimum cell number that can be detected was 1.72 × 10^6^ CFU/mL for *E. coli* Xen14, as compared to 3.58 × 10^7^ CFU/mL for *S. aureus* Xen29. 

### 3.4. Screening of a Soil Sample

To test and validate the performance of the BLSA assay, it was used to discover bacterial soil isolates with antimicrobial activity. A total of 264 isolates spanning over the wells of three plates were tested in triplicate using the BLSA assay in the agar-broth format. Figure 9 displays all the plate results during the validation test runs, with highlighted wells containing the putative antimicrobial-producing isolates or bacterial hits. In total, ten bacterial hits were identified of which seven were active against only *S. aureus*. The assay was generally repeatable with five bacterial hits displaying activity in all repeats, three in two out of three repeats, and two only in one repeat. Three of these isolates showed activity toward *E. coli*. There was a slightly bigger discrepancy in repeatability towards *S. aureus,* most likely to its lower maximum luminescence, higher LOD as compared to *E. coli*, and higher hit rate (sensitivity).

Upon closer investigation, it was established that the unrepeatable isolates resulted in reporter luminescence generally closer to the AbT, as compared to repeatable active isolates (Figure 10). A moderate decrease in the AbT would remove the “unrepeatable” isolates as bacterial hits and, therefore, increase repeatability. In Figure 10, it should be noted that there was a generally good distinction between active and non-active isolates towards the *E. coli* Xen14 reporter bacterium. Differentiation between positive and negative hits for *S. aureus* Xen29 was less clear, again, most likely due to its lower maximum luminescence and higher LOD. 

The activity of the bacterial hits identified in the BLSA assay was confirmed by the traditional simultaneous antagonism assay. From Table 2, as compared to the simultaneous antagonism assay, the BLSA assay gave more positive hits. However, this increased hit rate only occurred for the *S. aureus* Xen29 reporter bacterium. This could point either to a higher sensitivity of the reporter strain in broth than on agar to antimicrobial compounds, or to misidentification. The misidentification could also be in the traditional simultaneous antagonism assay, with antimicrobial compound not diffusing through the agar, limiting the detection of antimicrobial activity. Whatever the case, activity determination against *S. aureus* Xen29 would benefit from increased luminescence. Isolates active against *E. coli* Xen14 were all identified as such by the classical simultaneous antagonism assay and the BSLA assay. In total, the BLSA assay identified ten putative active isolates of which the activity of five isolates was confirmed by the classical simultaneous antagonism assay. Therefore, the BLSA resulted in a 96.2% reduction in cultures that needed to be screened, greatly reducing downstream analysis.

## 4. Discussion

The main goal of this study was to establish a natural antibiotic screening method that is simple, requiring only general laboratory instrumentation, yet compatible with automated HTS systems. The BSLA assay is based on the simultaneous bacterial antagonism assay. Unlike the well-known agar overlay assay of Waksman [11], where the activity of bacterial isolate extracts is determined against a reporter bacterium, the simultaneous antagonism assay directly tests the antimicrobial activity of an isolated bacterium during co-cultivation with a reporter bacterium. This has the advantage of circumventing culture extraction and partial purification, thereby reducing method complexity. This eliminates the need for extraction and purification optimization during the initial discovery of microbially produced antimicrobials. Furthermore, antimicrobials produced only during competitive co-culture would still be detected [30]. In the conventional simultaneous antagonism assay, antibacterial activity is observed in Petri dishes as a clear zone in the bacterial lawn surrounding the bacterial colony of interest. However, to conform to HTS, the assay must be microplate based, either utilizing the standard 96-well or 384-well formats. The inherent consequence of miniaturizing the conventional screening approach to a microtiter format is that clear zones in bacterial growth can no longer be used as an indicator of antibacterial activity due to the small size of wells. Furthermore, clear zone detection is mostly reliant on visual inspection, making it an unattractive prospect for HTS. Bioluminescent reporter bacteria are, therefore, ideal to circumvent both these issues due to their direct link to bacterial metabolism and simple detection by a luminometer. During the development of this assay, fluorescently tagged reporter bacterial strains were also considered; however, they were eventually excluded due to the generally greater background noise of fluorescence, large variability, and high misidentification rates.

The BLSA assay was designed to provide a binary answer as to whether bacterial isolates produce antimicrobial compounds. Therefore, the most important parameter to be determined for binary classification is the threshold above which bacterial isolates are considered inactive referred to as the AbT. We suggest that the AbT value be calculated as four-fold the standard deviation plus the mean of wells containing no reporter bacteria, which resulted in an AbT of 43 counts/s in our system. The AbT explained most of the variation in active and inactive known test bacteria. Most of the misclassification of known test bacteria was due to false negatives. This can be circumvented by increasing the AbT. However, we found that increasing the AbT results in a little gain in assay accuracy. The BLSA assay is meant to be applied as an initial screening assay in large screening pipelines. Therefore, false negatives are preferred above false positives which result in increased downstream processing. Furthermore, it was seldom that all repeats of a test bacterium were falsely classified as inactive. Assuming isolate selection is conducted at random, several wells with colonies of an active isolate (bacterial hit) will be present during screening, therefore reducing the chance of total loss due to misclassification as inactive. 

The BLSA assay was found to be very robust against reporter medium composition, allowing a broad range of culturing media. Therefore, as long as the reporter bacterium has sufficient luminescence in the desired reporter medium, the classification should be unaffected. Furthermore, we found increasing incubation time following reporter bacterium addition can result in more accurate classifications. This is likely due to slow-growing or low-producing test bacteria only producing sufficient amounts of its antibacterial compound during extended periods of growth. A further testament to assay robustness is the resistance of the reporter bacteria *E. coli* Xen14 and *S. aureus* Xen29, reducing pH to as low as 4, making it possible in the testing of most acid-producing bacteria. Although tolerant to reporter medium composition, increased misclassification was found in semi-solid reporter media. This is due to what is referred to as the halo effect. The halo effect is defined as surviving reporter bacteria surrounding an active test bacterium colony inside a well, resulting in a luminescent ring (halo) on the edges of the well. This results from hydrophobic antibacterial compounds produced by the test bacterium which are unable to diffuse throughout the entire well. Therefore, we suggest avoiding the usage of semi-solid reporter media.

Apart from simplicity, another goal was the scalability of the assay to an HTS method. Upon investigation, we found that the assay obtained Z’-factors above 0.5, except for the *E. coli* Xen14 reporter at 48 h of incubation, which is generally regarded as sufficient for HTS methodologies [25,31]. In fact, due to the binary nature of the BLSA assay, a Z’-factor of greater than zero is suggested. The S/R of greater than 10^4^ and 10^3^ for *E. coli* Xen14 and *S. aureus* Xen29, respectively, are magnitudes greater than the suggested minimum S/R of 10. These observations clearly demonstrated the compatibility of the BLSA assay with a high-throughput format. 

To test the real-world performance, the BLSA assay was used to test the activity of a total of 264 random bacterial soil isolates. This resulted in the identification of ten active isolates of which five were confirmed to be active by the traditional simultaneous antagonism assay. Thus, a total of five false positives resulted from the BLSA assay and all were from the *S. aureus* Xen29 reporter bacterium. This is likely due to the relatively lower maximum luminescence and higher LOD of *S. aureus* Xen29 as compared to *E. coli* Xen29 resulting in the overshadowing of luminescence by nutritional competition. To circumvent false negatives and decrease sensitivity, we propose the use of *S. aureus* strains with greater luminescence. Although several isolates were possibly misclassified, this is still a successful result, decreasing the possible pool of bacterial isolates to investigate during downstream processes by >95%.

## 5. Conclusions

In conclusion, we have developed a simple yet powerful preliminary screening assay for the identification of antibacterial-producing bacterial isolates. The assay is robust and easy to interpret and can be converted into an HTS pipeline with minimal effort.

## Figures and Tables

**Figure 1 microorganisms-10-02235-f001:**
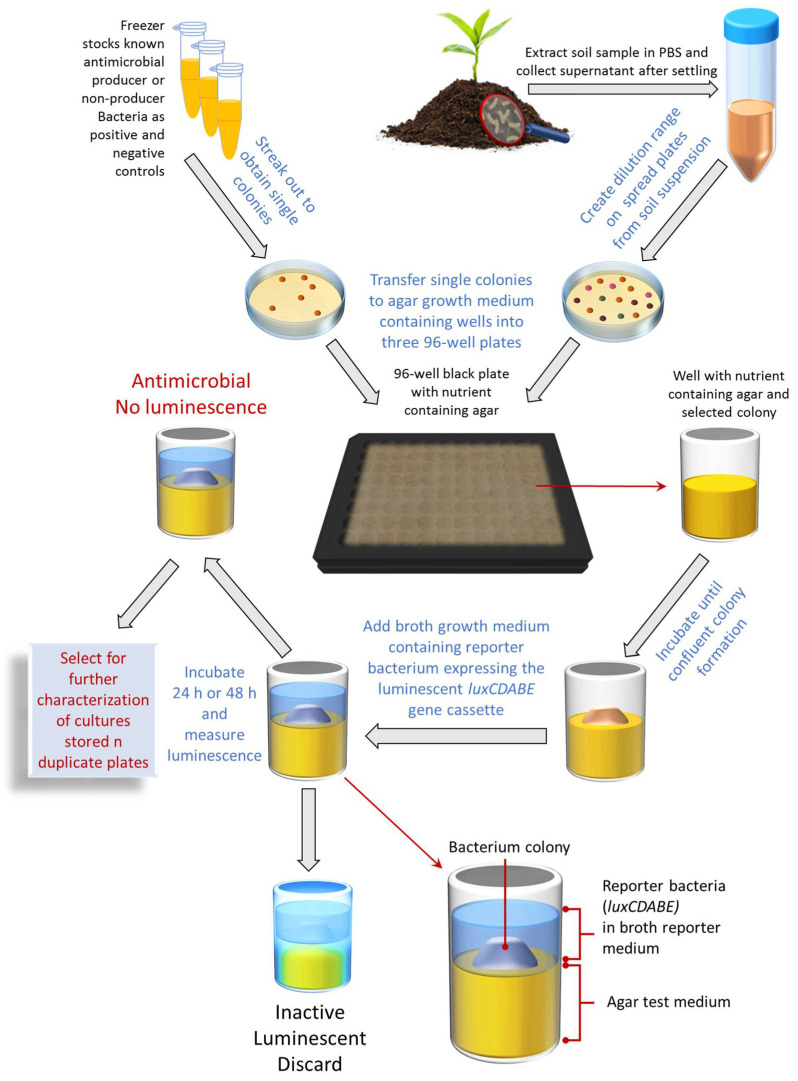
A schematic flow diagram detailing the steps in the BLSA assay for identifying antimicrobial bacterial producers from environmental samples.

**Figure 2 microorganisms-10-02235-f002:**
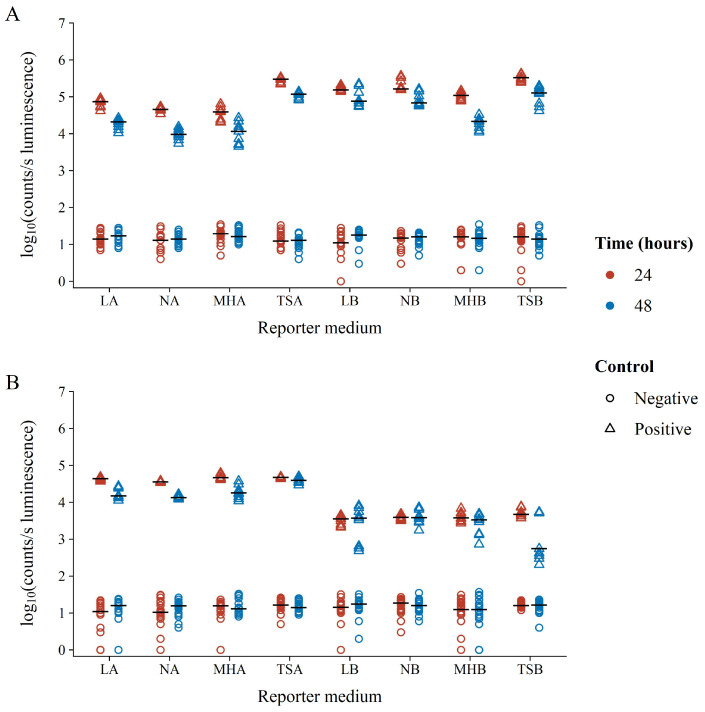
Log_10_ luminescence of negative (circles) and positive (triangles) BLSA assay controls in various reporter media following 24 (red) or 48 (blue) hours of growth. Positive controls consisted of wells containing suspensions of the reporter bacterium (**A**) *E. coli* Xen14 or (**B**) *S. aureus* Xen29 in the respective reporter medium. Negative controls contained reporter medium without the addition of any reporter bacteria.

**Figure 3 microorganisms-10-02235-f003:**
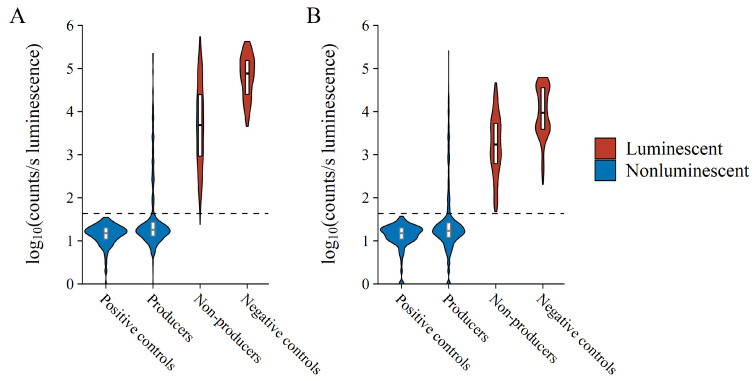
Collective Log_10_ luminescence of (**A**) *E. coli* Xen14 and (**B**) *S. aureus* Xen29 following test bacterium co-culture. Each x-axis variable contains observations from each experiment performed, thus collectively referring to all observations from all reporter media after both 24 and 48 h of incubation. Violin width is indicative of data point density. The horizontal dashed line indicates the AbT.

**Figure 4 microorganisms-10-02235-f004:**
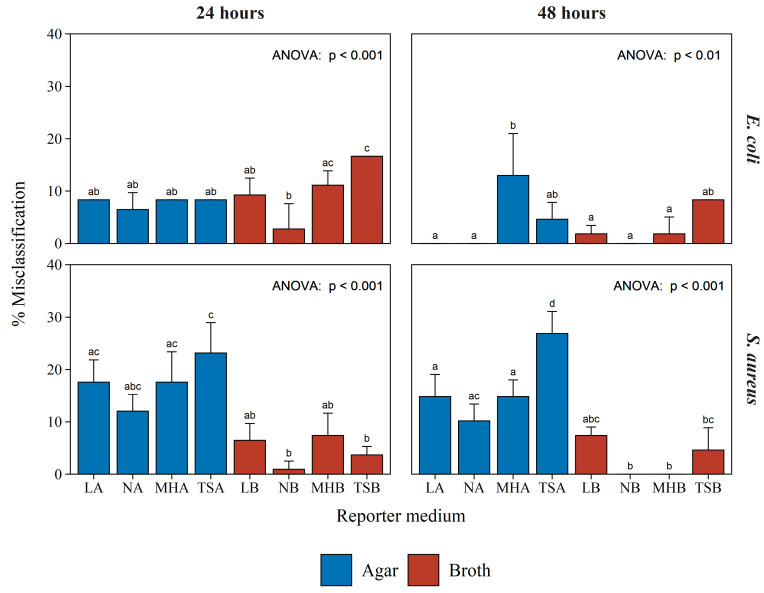
Misclassification rate of the BLSA assay while classifying 12 test bacteria in various reporter media. The top two graphs and bottom two graphs, with those in the left column at 24 h incubation and those in the right column at 48 h incubation, represent the misclassification rate when the reporter bacterium was *E. coli* Xen14 or *S. aureus* Xen29, respectively. Each bar shows the mean of three biological repeats with the error bar indicating the standard deviation. Significance, shown with corresponding letters (a, b, or c) was established by one-way ANOVA followed by the Tukey posthoc test.

**Figure 5 microorganisms-10-02235-f005:**
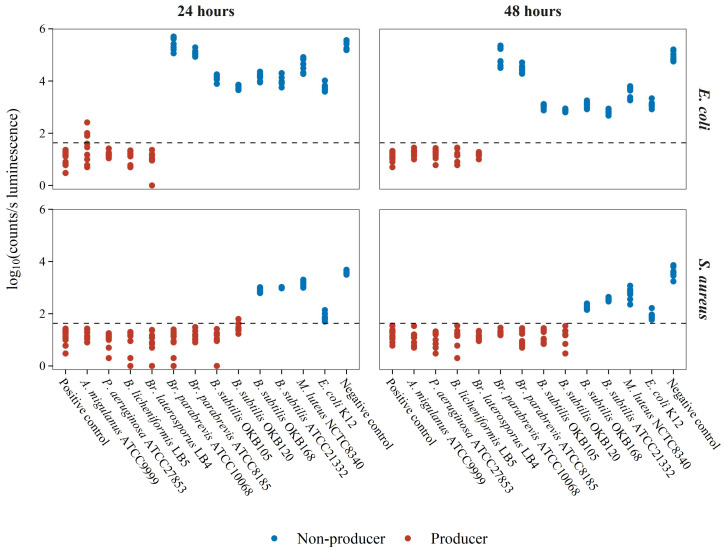
Luminescence of reporter bacteria following test-bacterium co-culture in the NB reporter medium. Luminescence is shown as log_10_(counts/s). The rows of the grid plot are representative of the reporter bacterium whereas columns represent incubation time. The horizontal dash line is indicative of the calculated AbT values. The data show each of the nine analyses for the 12 test bacteria. Negative controls contained only reporter bacteria whereas positive controls consisted of only uninoculated growth media.

**Figure 6 microorganisms-10-02235-f006:**
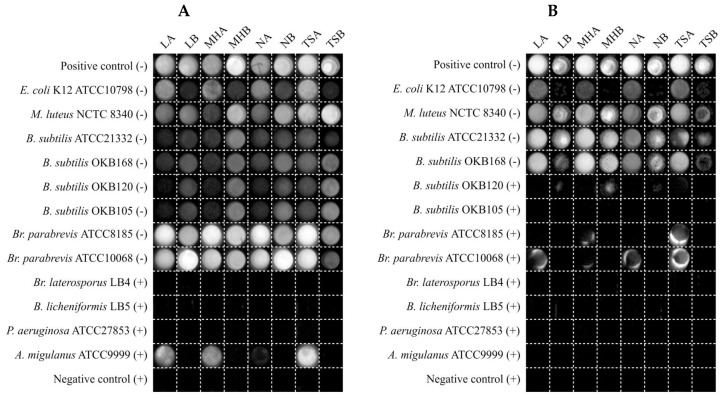
Chemiluminescent visualization of reporter bacteria (**A**) *E. coli* Xen14 and (**B**) *S. aureus* Xen29 following 24 h of test-bacterium co-culture in various reporter media. A (+) sign next to test bacterium names is indicative of an expected producer of an antimicrobial compound(s) targeting the reporter bacterium, whereas a (−) sign represents an expected non-producer.

**Figure 7 microorganisms-10-02235-f007:**
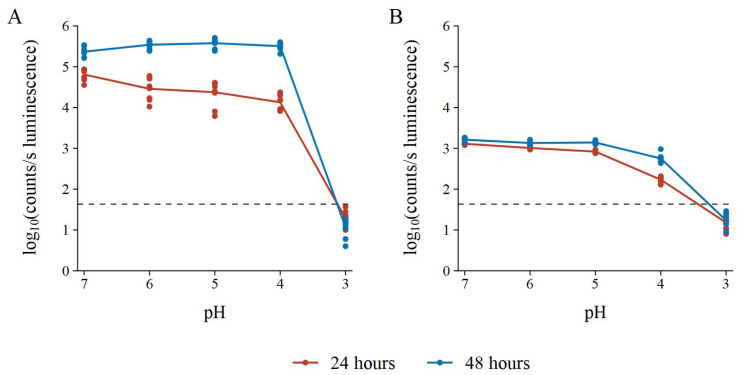
pH resistance of the BLSA assay with (**A**) *E. coli* Xen14 and (**B**) *S. aureus* Xen29 as reporter bacteria in the NB reporter medium. AbT is indicated by the horizontal dashed line. Luminescence is shown as log_10_(counts/s). The assay consisted of three biological repeats each with three technical repeats.

**Figure 8 microorganisms-10-02235-f008:**
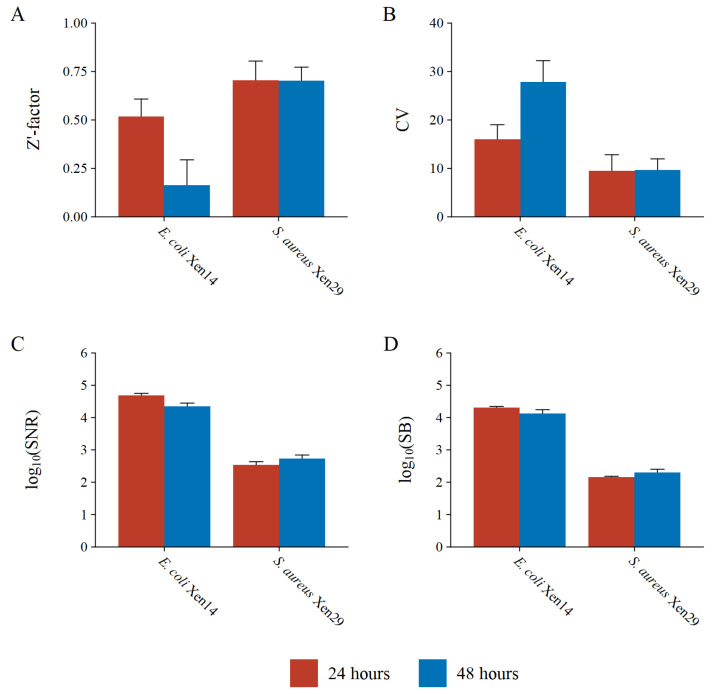
High-throughput assay quality measures including the (**A**) Z’-factor, (**B**) coefficient of variation of the maximum signal (negative controls), (**C**) signal-to-noise ratio, and (**D**) signal-to-background ratio for the reporter bacteria *E. coli* Xen14 and *S. aureus* Xen29 at 24 and 48 h of incubation.

**Figure 9 microorganisms-10-02235-f009:**
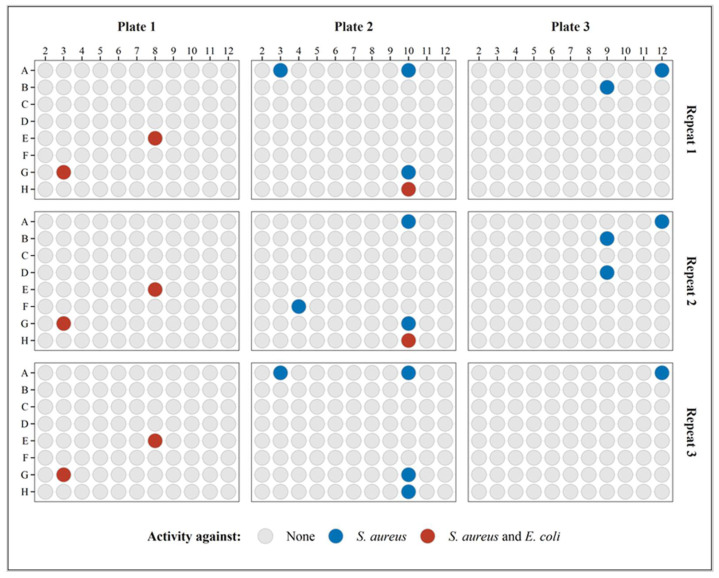
The activity of soil isolates is indicated in the respective wells and plates in triplicate. The wells with bacterial hits are indicated in color.

**Figure 10 microorganisms-10-02235-f010:**
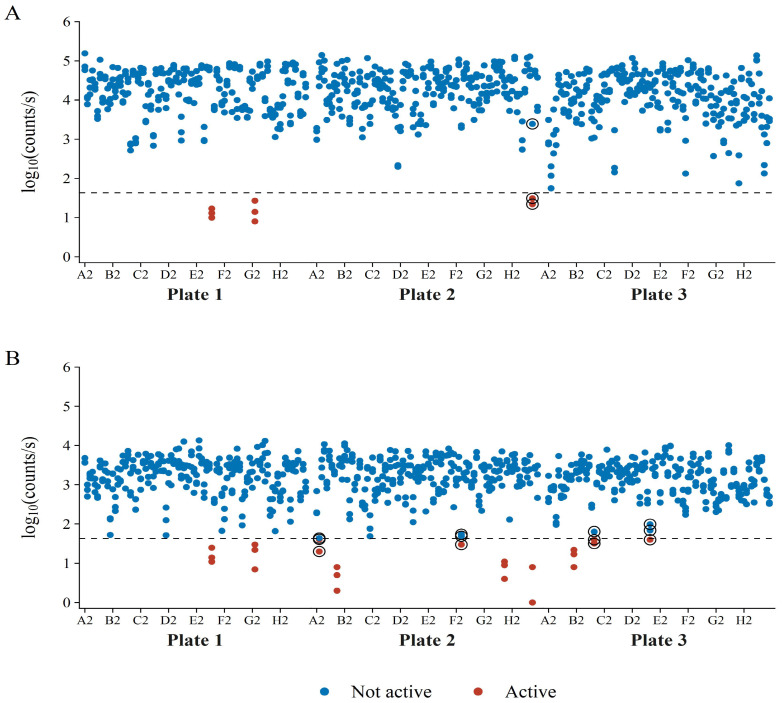
Luminescence of the reporter bacteria (**A**) *E. coli* Xen14 and (**B**) *S. aureus* Xen29 during co-culture with bacterial soil isolates. Bacterial hits are shown in red and non-active isolates appear in blue. The encircled data points show isolates that had borderline activity and have shown variable activity in the repeat analyses. Each isolate’s luminescence was measured in triplicate.

**Table 1 microorganisms-10-02235-t001:** Test bacteria in this study with their identified/putative antimicrobial compounds and their associated expected activity.

Bacterium	Strain	Activity ^a^		Antibacterial Compound(s) in Culture Extracts ^b^
		*E. coli*	*S. aureus*	
*Aneurinibacillus migulanus*	ATCC9999	+	+	Gramicidin S
*Bacillus licheniformis*	LB5	+	+	Bacitracin, Unknown
*Bacillus subtilis*	ATCC21332	-	+/-	Surfactin
*Bacillus subtilis*	OKB105	-	+	Surfactin
*Bacillus subtilis*	OKB120	-	+	Shortened surfactin ^c^
*Bacillus subtilis*	OKB168	-	-	None
*Brevibacillus laterosporus*	LB4	+	+	Loloatins and bogorols
*Brevibacillus parabrevis*	ATCC10068	-	+	Tyrothricin
*Brevibacillus parabrevis*	ATCC8185	-	+	Tyrothricin
*Escherichia coli*	K12	-	-	None
*Micrococcus luteus*	NCTC8340	-	-	None
*Pseudomonas aeruginosa*	ATCC27853	+	+	Rhamnolipids ^c^

^a^ Activity as determined by the simultaneous antagonism assay; refer to discussion below. ^b^ Refer to Appendix A; ^c^ too low level/resolution for direct injection detection by mass spectrometry.

**Table 2 microorganisms-10-02235-t002:** BLSA and simultaneous antagonism activity of active bacterial soil isolates identified by the BLSA assay.

Plate	Well	*E. coli* Xen14		*S. aureus* Xen29	
		BLSA	SA	BLSA	SA
1	E8	+	+	+	+
1	G3	+	+	+	+
2	A3	-	-	+	+
2	A10	-	-	+	+
2	F4	-	-	+	-
2	G10	-	-	+	-
2	H10	+	+	+	+
3	A12	-	-	+	-
3	B9	-	-	+	-
3	D9	-	-	+	-

## Data Availability

All data are curated and available from authors by direct request.

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
