# Peer review of "Direct Detection of Antibacterial-Producing Soil Isolates Utilizing a Novel High-Throughput Screening Assay"

_microorganisms, 2022, doi:10.3390/microorganisms10112235_

Round 1

Reviewer 1 Report

Laubscher and Rautenbach have wrote a nice paper in which they  propose an original method for screening environments for the presence of antimicrobial-producing bacteria so-called bioluminescent simultaneous antagonism assay (BLSA) based on previous knowledge but that may be very useful for large throughput screening of antimicrobial producing bacteria. Moreover it should be emphasized that new natural antimicrobial products are scarce and that in spite the good perspectives of chemical synthesis or chemical modifications  of already known products, the search for new natural ones is a strategy that also should be promoted. The introduction is well written and informative and introduces elegantly the topic. A few minor changes are listed below.

L 40 hopital …..hospital

L 48-49 antibiotic----antimicrobial (Salvarsan)

L50 Staphylococcus aureus in italics. Then, in  the rest of the manuscript. All scientific names of genera and species should be in italics (in some sections are well typed but not in others e.g bacterial strains section).

L70 cell membranes….bacterial envelopes

Materials and methods section is in general well written and allow the repetition of the experiments, nevertheless a few improvements would be interesting.

For instance authors wrote “Assay plates were subsequently inoculated with test bacteria, grown as described previously, by transferring a single colony per

well from streak plates using a sterile 200 μL pipette tip

It is not clear how they transfer a colony with a tip pipette, is the colony suspended? (and subsequently will grow covering the surface of test medium) or is deposited in a limited zone as a semisolid and the area depends upon the ability of the bacterium to move on the soft agar?

Also the reason(s) for a such a long incubation period (7 days) for bacteria such as B. subtilis or P. aeruginosa merited to be commented. (it seems to me that after 7 days incubation the proportion of dead P. aeruginosa     or B. subtilis  individuals “may cause”(?) false positives, but I’ve not evidence of this). As far as I know the tested bacteria are fast growing microorganisms that means that secondary metabolites (and thus, antimicrobials) may appear before 48 hours incubation. Why a so long incubation?. In general this merits some comment.

Results are well presented and provide knowledge to write the discussion. I’ve no comments on this.

Thus, I believe the paper merits publication in Microorganisms journal after introducing the minor changes suggested and maybe a few sentences  on the suggested questions.

Author Response

Thank you for your time and effort with our manuscript. Refer to the attached file for of detailed response and changes to the manuscript.

Reviewer 2 Report

This paper proposes a luminescence assay to screen for bacteria producing novel antimicrobials from soil. The proposed bioluminescent simultaneous antagonism assay utilizes luminescent Escherichia coli (Gram-negative bacteria) and Staphylococcus aureus (Gram-positive bacteria) as reporter bacteria, and has the potential to search for useful bacteria which from a wide variety of soils. This assay has the potential to identify useful antibiotic-producing bacteria from a wide variety of soils. This assay is expected to be developed in the future and is considered highly original. However, the assay procedure and evaluation method are complicated and not well explained, making it difficult to understand. In addition, there is a lack of discussion and output regarding screening methods from actual soils and methods for isolating target bacteria. A schematic diagram of the proposed method should be presented in an easy-to-understand manner. The experimental method is complicated, and the operations, procedures, and evaluation methods using well plates are very difficult to understand. Since this study is a proposal for a screening method, please provide a flow diagram that clearly shows the experimental operation, the control of luminescence intensity, and the data and handling of the test.

Specific points raised are as follows:

1) 2.1. Bacterial strains:

Reasons for selection of bacteria for testing and explanation of antibiotic producers and non-producers are needed. It is easier to understand if you create a Table like Table 1 that includes information on test bacteria. Reasons for selection of reporter bacteria E. coli Xen14 and S. aureus Xen29 are also needed.

2) The sections 2.3.–2.7. have the same text describing the analysis. “All analysis were done in the R statistical software [23] where the tidy verse R packages were used for data preparation and visialization [24].”

Since each section has different objectives and evaluation items, the purpose of evaluation by R statistical analysis and the analysis method should be specifically indicated.

3) Table 1:

Why does the same species of Bacillus subtilis produce the same antimicrobial surfactin but with different activity against E. coli and S. aureus?

4) The assay results are compared between agar and broth. As a result, agar seems to be more suitable for the assay. However, when assaying actual soil samples, the conditions are completely different from those of agar, and the effect of luminescence, i.e., the evaluation results when assaying agar and soil samples, has not been examined. Will the sensitivity of the luminescence evaluation in the assay of soil samples be reduced?

5) Soil samples contain a great variety of substances that affect luminescence and fluorescence, such as humic substances. Will luminescence-affecting substances in the soil affect the assay?

6) The paper as a whole contains some typographical errors and unclear sentences. The revised manuscripts should be thoroughly polished.

7) Figures and diagrams should be in color for easier understanding.

Author Response

(The authors gave the same response as above.)

Reviewer 3 Report

The manuscript by Laubscher and Rautenbach describes the validation and utilization of a high-throughput screening assay for identification of novel antimicrobials. Clearly there is a need to identify new classes/representatives of antibiotics and the approach used here should streamline such identification. The method relies on screening via co-culture with either Staph aureus (lux gene containing) or E. coli with the same bioluminescent reporter (lux).

Scientific Points

1.       L33 This is overstated and overly dramatic. To say that “most cases of a trivial infection” would certainly result in death seems unlikely. Common colds, mild diarrheas would have been common and self-limited.

2.       L50, Section 2.1, L502, L506. Please italicize all taxa throughout manuscript.  

L80 Perhaps use “full attention” or “extensive attention” since marine drugs have certainly been identified (there is an MDPI journal devoted to this topic).

3.       L92, L95 (and other examples if present)—italicization of gene names.

4.       L97 The authors include refs 19-22 as examples of lux being used. I was readily able to find several examples where S. aureus/lux constructs have been used for antibiotic detection, so these should be included (and any using E. coli). Also, since the title indicates that this is a novel screening approach, the Introduction should include what the limitations of the previous citations are and the Discussion should include a comparison or clear discussion of how the system described herein, is novel/advantageous.

5.       More details about the constructs are required. Which promoter? Chromosomal? Plasmid? Previously used?

6.       L236 The colonies were selected at random—so did this include fast and slow growers, small and large colonies, embedded colonies that were presumptive actinomycetes?

7.       Last line of Fig 5 legend. This is a little misleading—perhaps indicate that it is predicted to be active against the particular reporter (otherwise the labelling of +/- differs in each panel.

8.       L532 I do not think that the authors mentioned in the text that the producers made peptide antibiotics.

Language points/suggestions

1.       Please use continuous line numbering throughout. It pauses p9-12.

2.       L12 has yet

3.       L60, L418 “Selman” can be deleted on 2nd-3rd use of Waksman name

4.       L63 decreased

5.       L65 Introduces HST abbreviation here and then use L90, 417,427,433,473

6.       L114-6 can be deleted since t is repeated L118-121 and L125

7.       L107 and L129 Please use one format for States in USA

8.       L128, L135, L139, L145 (and add aliquot), L163, L230 (two examples), L237, L429 In these examples, the numbers are typically followed by hyphens

9.       L143 Define CFU on first use

10.   L152 The location of Thermo has already been given

11.   L155, L179, L192, L209, L225. These two lines are repeated throughout and could be placed as a final section on statistics. In addition, it should be “All analyses were” (in each example)

12.   L172 (two instances) This should be Mueller or Muller with the umlaut on the “u”

13.   L213 medium

14.   217 was made

15.   Figure 2. It may be a PDF issue, but my copy lacks the “t” on Nonluminescent

16.   Bottom of p9 Please add a space after “using”

17.   Fig 3. Depending on journal style “post-hoc” might require italicization

18.   L332 The other “Figure” references are not bolded.

19.   L459 bacteria

20.   L518 additional punctuation can be removed after WEL and before resources

21.   L523 Please clarify M…

22.   L525 were funded

23.   L532 bacterial producers (insert space)

24.   L544 Arch. (to match L541)

25.   L560 A new

Author Response

(The authors gave the same response as above.)

Reviewer 4 Report

The manuscript by Laubscher and Rautenbach on the use of various assays to determine the antibacterial activity of select bacterial strains is well-written. It provides valuable information in the race against antibacterial resistance. I am sure the manuscript will be helpful for a wide range of readers from different biological domains. I believe the manuscript should be accepted in its current form. However, I have one question and have highlighted a few typos in the manuscript, which are as follows.

Question!

There are numerous probiotic strains that have been predicted to have antibacterial activity by various bioinformatic tools. I would like to ask why the specific strains were selected for this study. Did you do any analyses in determining the antibacterial potential of these specific strains? or was it according to the literature?

Minor comments!

In subsection 2.1 of Bacterial strains, I think the names of the strains should be italicized.  

Author Response

Thank you for your time and effort with our manuscript. Refer to the attached file for our detailed response and changes to the manuscript.

Round 2

Reviewer 2 Report

The revised manuscript has been corrected by properly reflecting the comments and suggestions. In particular, the introduction of assay flow charts has improved the value of articles. I recommend the publication of this paper.